# Designing Bio-Based Color Sensor from Myofibrillar-Protein-Based Edible Film Incorporated with Sappan Wood (*Caesalpinia sappan* L.) Extract for Smart Food Packaging

**Iis Rostini** [1,*] , **Junianto** [2] **and Endang Warsiki** [3]

1 Doctoral Program of Agriculture Science, Faculty of Agriculture, Universitas Padjadjaran, Sumedang 45363, Indonesia
2 Laboratory of Fisheries Processing Product, Department of Fisheries, Faculty of Fisheries and Marine Science, Universitas Padjadjaran, Sumedang 45363, Indonesia; junianto@unpad.ac.id
3 Department of Agroindustrial Technology, Faculty of Agricultural Engineering and Technology, Bogor Agricultural University, Bogor 16680, Indonesia; endangwarsiki@apps.ipb.ac.id
* Correspondence: iis.rostini@unpad.ac.id

**Abstract:** The use of intelligent sensor-based packaging in food products allows the quick evaluation of food quality. This study aimed to design a color sensor from surimi utilizing sappan wood extract (SWE) for intelligent food packaging. The myofibrillar-protein-based edible film incorporated the SWE at different concentrations (0.15%, 0.25%, and 0.35%). The physical properties of the sappan wood–surimi edible film (SSEF) were analyzed, and the color changes at various pH levels and soaking times under different conditions were assessed. According to the results, different concentrations of the SWE significantly affected the thickness, transparency, tensile strength, and elongation at break of the film. However, the values were still within the acceptable range. The pH values increased the film's darkness, redness, and blueness. Based on the evaluation of the SSEF under different soaking conditions, the color changes of the film in acidic conditions were more stable than in neutral and alkaline conditions. This study showed that SSEF can be used as intelligent food packaging. It can act as a color sensor due to its sensitivity to the changes in the pH condition of the product.

**Keywords:** natural dye; pH sensor; smart packaging; surimi-based; tilapia protein



## 1. Introduction

Fresh food goods can rapidly deteriorate due to the physical damage that might be sustained during the processes brought on by chemical reactions, enzyme activity, and microbial contamination [1]. The development of new packaging solutions is a result of the development of packaging technology in response to consumer expectations for food products that are fresh, safe, and of exceptional quality [2,3]. Food waste and food degradation can be prevented by using packaging as a storefront for information about the food's condition. Packaging can also actively contribute to the preservation of food. In this regard, research and development are underway on active and intelligent packaging [1]. Bioactive substances, such as antimicrobials [4] and antioxidants [5], are present in active food-packaging materials to increase the shelf lives of, preserve the quality of, and stabilize food-packaged commodities [6]. Furthermore, intelligent food-packaging solutions are designed to monitor the environment or the conditions of the food, identify any physical, chemical, or biological changes, and take appropriate action. The intended outcome is to quickly evaluate food quality [7–9].

There are three categories of intelligent sensor-based packaging material: freshness, time–temperature, and gas. These materials can be used to track inaccuracies in temperature changes along the supply chain, modifications in the gas composition in food packing,

and the deterioration of freshness due to variations in the amount of metabolites, which are indicative of microbial growth [3,10]. The most widely used freshness indicators include a stable base and a dye that responds to pH variations by changing color and providing a visual response to the environment inside the package [3,7]. Numerous studies have examined different synthetic dyes as pH indicators [11–13], but the leaching of the dye and consumer awareness of chemically manufactured dyes' negative effects raises questions due to their toxicity and bioaccumulation [3,7,14]. Due to their biodegradability, lack of toxicity, lack of carcinogenicity, and ecologically benign manufacturing process, natural dyes derived from a variety of sources are potential options [15,16]. The red naphthoquinone pigment, shikonin, from the roots of gromwell, is one example of a natural pigment used in colorimetric indicator systems. Other examples include anthocyanins from barberry (*Berberis vulgaris* L.), black carrots (*Daucus carota* L.), saffron (*Crocus sativus* L.), red cabbage (*Brassica oleraceae*), and saffron [8,9,17,18].

Sappan wood (*Caesalpinia sappan* L.) is naturally present across Asia, including Thailand, China, and Japan. Due to its many beneficial uses, this wood plant is now grown in several parts of the world, including Africa, Europe, North America, and South America. Since the wood is inexpensive and lacks a distinct flavor, it has the potential to be utilized as a natural red dye, as opposed to a synthetic dye [19]. The main active ingredient in sappan is brazilein, a white phenolic compound with two aromatic rings: one pyrone, and one five-membered ring. However, the brazilein structure's hydroxyl group is simply oxidized. It can transition into a carbonyl group, resulting in a structural alteration and the production of the vibrant chemical, brazilein. Brazilein is a polyphenolic chemical; hence, changes in pH are likely to cause the hydroxyl group in its molecule to change color [20,21]. In addition to brazilein, *C. sappan* is also thought to be a possible source of anthocyanins. These natural compounds may be candidates to replace synthetic dyes due to their attractive, vibrant hues (orange, red, and purple), which fade quickly when exposed to light, oxygen, high temperatures, pH, salt stress, and enzymes [22].

The incorporation of biopolymers in active packaging has attracted the attention of numerous researchers. These biopolymers include the units created by a covalent peptide link, proteins [23]. Many crucial protein sources can be obtained from various plant or animal sources. Due to the abundant resources in these genuine items, researchers started to extract polypeptides from diverse vegetable and animal products and by-products [24,25]. Surimi, a by-product of minced, deboned fish meat, contains concentrated myofibrillar fish proteins [26]. The attributes of myofibrillar-protein-based films are slightly superior to those of known protein films, and myofibrillar protein exhibited outstanding film-forming capacities in both acidic and alkaline environments [27–29].

Some researchers have performed bio-based sensor research [30,31]. However, no studies have been conducted on the changes in physical properties and colors of myofibrillar-protein-based edible films combined with sappan-wood extract at varied pH and soaking times under different conditions. Thus, the present research aims to investigate the physical properties of myofibrillar-protein-based edible films with the addition of different concentrations of sappan-wood extract. The films' colors at different pH and soaking conditions were also evaluated.

## 2. Materials and Methods

### 2.1. Materials

The surimi was obtained from the processing of Nile tilapia (*Orechromis niloticus*). It was bought from a supermarket (West Java, Indonesia), with the process performed following the method described by [32], with modifications. The sappan-wood extract was obtained from the water extraction of sappan-wood simplisia obtained from a local market (Central Java, Indonesia) [33]. All chemicals were analytical-grade.

## 2.2. Preparation of Sappan Wood Surimi Edible Film

The preparation of sappan wood–surimi edible film (SSEF) followed the method described by [32], with slight modifications. Frozen surimi (10% $w/w$) was thawed for 20 min and mixed with distilled water (150 mL) and 1 M HCl until the pH was 3. The mixture was homogenized using a homogenizer (PRO250 Homogenizer, Thomas 1204B63, Thomas Scientific, Swedesboro, NJ, USA) for 30 min at 55 °C, with 50% glycerol from the weight of the surimi ($w/w$). The mixture was then sieved using a 150-mesh nylon sieve. The sappan-wood extract (SWE) was added to the film mixture at different concentrations (0.15%, 0.25%, and 0.35% $w/v$) as a natural dye. The SSEF mixture was homogenized, poured onto a glass plate (20 × 20 cm), and dried in a hot-air oven (Binder, Binder GmbH, Tuttligen, Germany) at 50 °C for 48 h. After drying, the SSEF film was packed in a polyethylene bag and placed in a desiccator for further use.

## 2.3. Physical-Properties Analysis
### 2.3.1. Thickness

The SSFEs' thickness was measured using a digital micrometer (Mitutoyo, Tokyo, Japan). The measurements were performed on different areas of the films.

### 2.3.2. Transparency

The film transparency (%) was measured by the spectrophotometric method using a UV spectrophotometer (model UV-160, Shimadzu, Kyoto, Japan) at 600 nm [34]. The following equation calculated the transparency value of the film:

$$\text{Transparency value} = (-\log \text{T}_{600})/\text{x} \tag{1}$$

where $\text{T}_{600}$ is the fractional transmittance at 600 nm, and x is the thickness of the film (mm).

### 2.3.3. Mechanical Properties

The tensile strength (TS) and elongation at break (EAB) of the SSEFs were measured according to the standard protocols [35], with some modifications. Films were cut into strips (1 cm × 10 cm) and kept in a desiccator containing NaBr solution with RH of 57% for 72 h prior to the test. The measurement was performed using a texture analyzer (Stable Microsystem, Surrey, UK) calibrated with a 5 kg load before use. Prior to the analysis, both ends of the film strips were marked. The initial separation distance and the velocity were fixed at 40 mm and 0.40 mm/s, respectively. A 300 mm/min pre-test speed and a 600 mm/min post-test speed were automatically set as the trigger forces used for the analysis. The cell-load capacity and the return distance of the texture analyzer were 30 kg and 190 mm, respectively [36].

## 2.4. Color Values at Different pH and Soaking Times at Different Conditions

The color (L*, a*, and b*) values of the SSFEs at different pH (1 to 14) were obtained using a buffer solution at respective pH and soaking times (0–20 min, observed every 2 min) under different conditions (acid, neutral, and alkaline). The color was measured using a Colorimeter (ColorFlex, Hunter Lab Inc., Reston, VA, USA), with the L* representing the dark-light spectrum, the a* representing red intensity, and the b* representing yellow intensity [37]. The total color difference (ΔE) was calculated as follows:

$$\Delta\text{E} = \sqrt{((\Delta\text{L*})^2 + (\Delta\text{a*})^2 + (\Delta\text{b*})^2)} \tag{2}$$

where ΔL*, Δa*, and Δb* were the differences between the color parameters of the control sample and the color parameters obtained at different pH and soaking times under different conditions.

### 2.5. Statistical Analysis

Each result was verified three times, and it was given as mean SD (standard deviation). Duncan's multiple-range test (DMRT) and SPSS software version 17 (SPSS Inc., Chicago, IL, USA) were used for the statistical analysis, with a 95% confidence level selected for significance.

## 3. Results

### 3.1. Physical Properties

The results for the physical properties of the SSEFs are shown in Table 1. The thickness of the SSEFs ranged between 0.17 and 0.22 mm. Based on the results, the transparency of the SSEFs ranged between 0.84 and 2.16. The determination of the TS and the elongation at break of the SSEF revealed that the values of these properties ranged between 7.70 and 10.15 mPa and between 12.68% and 15.70%, respectively. The addition of the SWE at different concentrations in the formulation of the SSEF significantly ($p < 0.05$) affected the properties of the resulting film.

**Table 1.** Physical properties of SSEF with different SWE concentrations.

| Treatment (%) | Thickness (mm) | Transparency | Tensile Strength (mPa) | Elongation at Break (%) |
|---|---|---|---|---|
| 0.00 | 0.13 ± 0.01 [d] | 3.46 ± 0.01 [d] | 15.43 ± 0.01 [a] | 11.23 ± 0.01 [c] |
| 0.15 | 0.17 ± 0.01 [c] | 2.16 ± 0.13 [a] | 10.15 ± 1.1 [b] | 12.68 ± 1.17 [b] |
| 0.25 | 0.19 ± 0.01 [b] | 1.26 ± 0.03 [b] | 8.48 ± 1.0 [c] | 14.80 ± 0.96 [a] |
| 0.35 | 0.22 ± 0.01 [a] | 0.84 ± 0.04 [c] | 7.70 ± 0.7 [c] | 15.70 ± 1.26 [a] |
| *p*-value | <0.0001 | <0.0001 | <0.0001 | <0.0001 |

Remarks: Data are presented as mean ± SD. A different superscript letter indicates a significant difference ($p < 0.05$) in treatment.

### 3.2. Color Values of SSEF at Different Conditions

3.2.1. Color Values at Different pH

The sensitivity of the SSEFs with varying SWE concentrations to the pH variations was tested by immersing them in buffer solutions with pH values ranging from 1 to 14 (Table 2). Based on the color measurement at the same pH value, the L* of the SSEF with 0.15% SWE exhibited a significantly ($p < 0.05$) higher value (65.86–74.97) than the films with 0.25% (64.75–73.56) and 0.35% (44.35–60.96) SWE. Conversely, the a* and b* values of the SSEF containing the highest concentration of SWE showed significantly ($p < 0.05$) higher values (16.72–58.25 and 25.14–43.58, respectively) than the SSEFs with 0.25% (7.68–50.22 and 15.65–29.87, respectively) and the 0.15% (5.43–48.87 and 11.75–22.15, respectively) extract (Table 1). Consequently, the total color difference (ΔE) of the SSEFs with the addition of 0.15%, 0.25%, and 0.35% SWE increased significantly (2.30–44.20, 8.37–43.90, and 3.77–45.86, respectively; $p < 0.05$).

**Table 2.** Apparent color and colorimetric parameters (L*, a*, and b*) of SSEF with different SWE concentrations at different pH values (1 to 14).

| Treatment | | pH | | | | | | | | | | | | | |
|---|---|---|---|---|---|---|---|---|---|---|---|---|---|---|---|
| | | 1 | 2 | 3 | 4 | 5 | 6 | 7 | 8 | 9 | 10 | 11 | 12 | 13 | 14 |
| 0.15% | L* | 74.97 ± 0.02 Aa | 74.83 ± 0.02 Ba | 74.28 ± 0.03 Ca | 73.96 ± 0.02 Da | 73.55 ± 0.02 Ea | 73.29 ± 0.02 Fa | 72.96 ± 0.01 Ga | 72.45 ± 0.02 Ha | 72.03 ± 0.02 Ia | 71.85 ± 0.02 Ja | 71.25 ± 0.01 Ka | 70.49 ± 0.03 La | 69.22 ± 0.03 Ma | 65.86 ± 0.02 Na |
| | a* | 5.43 ± 0.01 Nc | 6.75 ± 0.04 Mc | 8.29 ± 0.02 Lc | 9.45 ± 0.07 Kc | 9.86 ± 0.02 Jc | 10.39 ± 0.01 Ic | 10.67 ± 0.01 Hc | 11.33 ± 0.01 Gc | 12.24 ± 0.01 Fc | 16.33 ± 0.01 Ec | 19.51 ± 0.04 Dc | 34.25 ± 0.02 Cc | 47.28 ± 0.03 Bc | 48.87 ± 0.02 Ac |
| | b* | 22.15 ± 0.02 Ac | 21.85 ± 0.03 Bc | 21.22 ± 0.03 Cc | 20.26 ± 0.01 Dc | 19.89 ± 0.03 Ec | 19.04 ± 0.01 Fc | 18.74 ± 0.01 Gc | 18.18 ± 0.01 Hc | 17.84 ± 0.02 Ic | 16.72 ± 0.04 Jc | 15.68 ± 0.02 Kc | 13.39 ± 0.01 Lc | 12.46 ± 0.01 Mc | 11.75 ± 0.03 Nc |
| | ΔE | 2.30 ± 0.02 Lc | 1.93 ± 0.01 Mc | 2.33 ± 0.01 Lc | 3.40 ± 0.07 Kb | 3.81 ± 0.03 Jb | 4.62 ± 0.01 Ic | 5.01 ± 0.01 Hc | 5.88 ± 0.01 Gc | 6.89 ± 0.02 Fc | 11.04 ± 0.02 Ec | 14.42 ± 0.04 Dc | 29.13 ± 0.01 Cc | 42.14 ± 0.02 Bb | 44.20 ± 0.01 Ab |

**Table 2.** *Cont.*

| Treatment | | 1 | 2 | 3 | 4 | 5 | 6 | 7 | 8 | 9 | 10 | 11 | 12 | 13 | 14 |
|---|---|---|---|---|---|---|---|---|---|---|---|---|---|---|---|
| | | | | | | | | pH | | | | | | | |
| 0.25% | |  | | | | | | | | | | | | | |
| | L* | 73.56 ± 0.01 Ab | 73.04 ± 0.05 Bb | 72.81 ± 0.02 Cb | 72.22 ± 0.02 Db | 71.74 ± 0.01 Eb | 71.55 ± 0.02 Fb | 71.11 ± 0.01 Gb | 70.86 ± 0.02 Hb | 70.53 ± 0.01 Ib | 70.18 ± 0.02 Jb | 69.57 ± 0.02 Kb | 67.85 ± 0.01 Lb | 64.92 ± 0.01 Mb | 64.75 ± 0.02 Nb |
| | a* | 7.68 ± 0.01 Nb | 8.12 ± 0.01 Mb | 9.55 ± 0.01 Lb | 10.89 ± 0.02 Kb | 11.58 ± 0.02 Jb | 11.95 ± 0.01 Ib | 12.32 ± 0.01 Hb | 12.77 ± 0.01 Gb | 13.98 ± 0.01 Fb | 18.32 ± 0.01 Eb | 21.75 ± 0.02 Db | 38.63 ± 0.02 Cb | 49.78 ± 0.01 Bb | 50.22 ± 0.01 Ab |
| | b* | 29.87 ± 0.01 Ab | 29.34 ± 0.04 Bb | 28.75 ± 0.03 Cb | 27.55 ± 0.02 Db | 26.98 ± 0.01 Eb | 25.54 ± 0.03 Fb | 24.66 ± 0.01 Gb | 22.78 ± 0.01 Hb | 21.97 ± 0.01 Ib | 21.01 ± 0.01 Jb | 19.56 ± 0.01 Kb | 17.45 ± 0.02 Lb | 15.92 ± 0.01 Mb | 15.65 ± 0.02 Nb |
| | ΔE | 8.37 ± 0.01 La | 8.18 ± 0.03 Ma | 8.10 ± 0.03 Na | 8.61 ± 0.03 Kba | 8.89 ± 0.03 Ja | 10.14 ± 0.02 Ia | 10.84 ± 0.01 Ha | 12.61 ± 0.02 Ga | 13.54 ± 0.02 Fa | 15.91 ± 0.02 Ea | 18.86 ± 0.02 Db | 32.78 ± 0.01 Ca | 43.38 ± 0.01 Ba | 43.90 ± 0.01 Ac |
| 0.35% | |  | | | | | | | | | | | | | |
| | L* | 60.96 ± 0.02 Bc | 59.45 ± 0.05 Cc | 58.97 ± 0.02 Cc | 58.24 ± 0.02 Ec | 57.55 ± 0.02 Ec | 57.21 ± 0.02 Fc | 56.75 ± 0.02 Gc | 55.65 ± 0.01 Hc | 54.17 ± 0.01 Ic | 53.67 ± 0.02 Jc | 51.11 ± 0.04 Kc | 48.25 ± 0.02 Lc | 44.86 ± 0.02 Mc | 44.35 ± 0.02 Nc |
| | a* | 16.72 ± 0.01 Na | 17.56 ± 0.04 Ma | 18.75 ± 0.03 La | 19.53 ± 0.01 Ka | 19.94 ± 0.02 Ja | 20.89 ± 0.02 Ia | 21.77 ± 0.02 Ha | 22.14 ± 0.01 Ga | 23.34 ± 0.03 Fa | 27.76 ± 0.02 Ea | 34.28 ± 0.01 Da | 45.73 ± 0.01 Ca | 55.25 ± 0.02 Ba | 58.25 ± 0.01 Aa |
| | b* | 43.58 ± 0.01 Aa | 43.25 ± 0.03 Ba | 42.63 ± 0.04 Ca | 42.21 ± 0.02 Da | 41.75 ± 0.03 Ea | 40.44 ± 0.01 Fa | 39.37 ± 0.01 Ga | 38.21 ± 0.03 Ha | 37.96 ± 0.02 Ia | 35.72 ± 0.01 Ja | 33.55 ± 0.02 Ka | 30.12 ± 0.03 La | 26.85 ± 0.02 Ma | 25.14 ± 0.02 Na |
| | ΔE | 3.77 ± 0.01 Lb | 3.03 ± 0.05 Nb | 2.83 ± 0.05 Mb | 3.26 ± 0.02 Kb | 3.98 ± 0.04 Jb | 5.36 ± 0.01 Ib | 6.69 ± 0.01 Hb | 8.28 ± 0.02 Gb | 9.65 ± 0.01 Fb | 13.61 ± 0.01 Eb | 20.31 ± 0.02 Db | 31.94 ± 0.02 Cb | 42.42 ± 0.02 Bb | 45.86 ± 0.02 Aa |

Remarks: Data are given as mean ± SD. A difference in uppercase superscript letter indicates a significant difference ($p < 0.05$) in soaking time. A difference in lowercase superscript letter indicates a significant difference ($p < 0.05$) in concentration.

### 3.2.2. Color Values at Different Soaking Times in Different Conditions

The SSEF with different SWE concentrations was also subjected to different soaking times under various conditions (Tables 3–5). Based on the color measurement of the SSEF under different soaking times in acidic conditions (Table 3), the L* of the SSEF with 0.15% SWE exhibited a significantly ($p < 0.05$) higher value (73.09–73.41) than the films with 0.25% (67.56–68.74) and 0.35% (59.75–60.62) SWE. Conversely, the a* and b* values of the SSEF containing the highest concentration of SWE showed significantly ($p < 0.05$) higher values (15.31–19.96 and 39.42–45.07, respectively) than the SSEFs with 0.25% (10.22–10.81 and 28.95–34.79, respectively) and 0.15% (5.65–6.29 and 19.61–21.15, respectively). The total color difference (ΔE) of the SSEF with the addition of 0.15%, 0.25%, and 0.35% SWE increased significantly (0.29–1.69, 0.26–5.99, and 1.49–7.37, respectively).

**Table 3.** Apparent color and colorimetric parameters (L*, a*, and b*) of the SSEFs with different SWE concentrations and soaking times in acidic conditions.

| Treatment | | 0 | 2 | 4 | 6 | 8 | 10 | 12 | 14 | 16 | 18 | 20 |
|---|---|---|---|---|---|---|---|---|---|---|---|---|
| | | | | | | | Time (min) | | | | | |
| 0.15% | |  | | | | | | | | | | |
| | L* | 73.09 ± 0.01 Ea | 73.09 ± 0.02 Ea | 73.16 ± 0.03 Da | 73.24 ± 0.02 Ca | 73.28 ± 0.01 Ba | 73.38 ± 0.01 Aa | 73.39 ± 0.01 Aa | 73.39 ± 0.02 Aa | 73.41 ± 0.02 Aa | 73.41 ± 0.02 Aa | 73.41 ± 0.01 Aa |
| | a* | 6.29 ± 0.02 Ac | 6.27 ± 0.01 Ac | 6.13 ± 0.02 Bc | 5.98 ± 0.01 Cc | 5.86 ± 0.03 Dc | 5.77 ± 0.02 Ec | 73.39 ± 0.01 Ac | 73.39 ± 0.02 Aa | 5.66 ± 0.01 Gc | 5.65 ± 0.03 Gc | 5.65 ± 0.01 Gc |
| | b* | 21.15 ± 0.01 Ac | 20.86 ± 0.02 Bc | 20.52 ± 0.01 Cc | 20.33 ± 0.02 Dc | 19.82 ± 0.01 Ec | 19.66 ± 0.02 Fc | 19.65 ± 0.02 Fc | 19.63 ± 0.02 FGc | 19.63 ± 0.01 FGc | 19.61 ± 0.03 Gc | 19.61 ± 0.01 Gc |
| | ΔE | 0 | 0.29 ± 0.02 Ec | 0.65 ± 0.02 Dc | 0.89 ± 0.02 Cc | 1.41 ± 0.02 Bc | 1.60 ± 0.02 Ac | 1.63 ± 0.02 Ac | 1.68 ± 0.02 Ac | 1.68 ± 0.01 Ac | 1.69 ± 0.02 Ac | 1.69 ± 0.01 Ac |
| 0.25% | |  | | | | | | | | | | |
| | L* | 67.56 ± 0.02 Gb | 67.56 ± 0.03 Gb | 67.68 ± 0.01 Fb | 66.23 ± 0.02 Db | 65.43 ± 0.01 Eb | 65.28 ± 0.01 Fb | 68.65 ± 0.01 Bb | 68.74 ± 0.03 Ab | 68.74 ± 0.01 Ab | 68.74 ± 0.02 Ab | 68.74 ± 0.03 Ab |
| | a* | 10.81 ± 0.02 Ab | 10.74 ± 0.02 Bb | 10.52 ± 0.01 Cb | 10.49 ± 0.02 Db | 10.31 ± 0.02 Eb | 10.27 ± 0.01 Fb | 10.23 ± 0.02 Gb | 10.23 ± 0.01 Gb | 10.22 ± 0.01 Gb | 10.22 ± 0.02 Gb | 10.22 ± 0.01 Gb |
| | b* | 34.79 ± 0.02 Ab | 34.54 ± 0.01 Bb | 33.48 ± 0.01 Cb | 32.27 ± 0.02 Db | 31.73 ± 0.01 Eb | 31.39 ± 0.01 Fb | 30.87 ± 0.01 Gb | 30.49 ± 0.01 Hb | 30.26 ± 0.01 Ib | 29.52 ± 0.01 Jb | 28.95 ± 0.01 Kb |
| | ΔE | 0 | 0.26 ± 0.01 Jb | 1.35 ± 0.01 Ib | 2.56 ± 0.02 Hb | 3.13 ± 0.01 Gb | 3.53 ± 0.01 Fb | 4.11 ± 0.01 Eb | 4.50 ± 0.02 Db | 4.72 ± 0.01 Cb | 5.43 ± 0.01 Bb | 5.99 ± 0.03 Ab |

**Table 3.** *Cont.*

| Treatment | | Time (min) | | | | | | | | | | |
|---|---|---|---|---|---|---|---|---|---|---|---|---|
| | | 0 | 2 | 4 | 6 | 8 | 10 | 12 | 14 | 16 | 18 | 20 |
| 0.35% | |  |  |  |  |  |  |  |  |  |  |  |
| | L* | 59.75 ± 0.01 Gc | 59.74 ± 0.02 Gc | 59.95 ± 0.01 Fc | 60.23 ± 0.01 Ec | 60.33 ± 0.02 Dc | 60.54 ± 0.02 Cc | 60.69 ± 0.02 Ac | 60.64 ± 0.02 Bc | 60.63 ± 0.02 Bc | 60.62 ± 0.03 Bc | 60.62 ± 0.01 Bc |
| | a* | 19.96 ± 0.02 Aa | 18.72 ± 0.01 Ba | 17.21 ± 0.02 Ca | 16.96 ± 0.03 Da | 16.58 ± 0.01 Ea | 15.42 ± 0.02 Fa | 15.37 ± 0.02 Ga | 15.34 ± 0.01 Ha | 15.31 ± 0.02 Ha | 15.31 ± 0.02 Ha | 15.31 ± 0.01 Ha |
| | b* | 45.07 ± 0.02 Aa | 44.25 ± 0.02 Ba | 42.23 ± 0.02 Ca | 42.21 ± 0.02 Ca | 40.05 ± 0.02 Da | 39.55 ± 0.02 Ea | 39.51 ± 0.02 Fa | 39.47 ± 0.01 Ga | 39.44 ± 0.02 GHa | 39.42 ± 0.01 Ha | 39.42 ± 0.01 Ha |
| | ΔE | 0 | 1.49 ± 0.02 Fa | 3.96 ± 0.01 Ea | 4.17 ± 0.02 Da | 6.08 ± 0.02 Ca | 7.19 ± 0.02 Ba | 7.27 ± 0.02 Aa | 7.31 ± 0.01 Aa | 7.35 ± 0.01 Aa | 7.37 ± 0.01 Aa | 7.37 ± 0.01 Aa |

Remarks: Data are given as mean ± SD. A difference in uppercase superscript letter indicates a significant difference ($p < 0.05$) in soaking time. A difference in lowercase superscript letter indicates a significant difference ($p < 0.05$) in concentration.

**Table 4.** Apparent color and colorimetric parameters (L*, a*, and b*) of the SSEFs with different SWE concentrations and soaking times in neutral conditions.

| Treatment | | Time (min) | | | | | | | | | | |
|---|---|---|---|---|---|---|---|---|---|---|---|---|
| | | 0 | 2 | 4 | 6 | 8 | 10 | 12 | 14 | 16 | 18 | 20 |
| 0.15% | |  |  |  |  |  |  |  |  |  |  |  |
| | L* | 73.06 ± 0.01 Aa | 73.01 ± 0.02 Ba | 72.97 ± 0.01 Ca | 72.63 ± 0.01 Da | 72.16 ± 0.02 Ea | 71.64 ± 0.03 Fa | 71.62 ± 0.01 Fa | 71.62 ± 0.01 Fa | 71.61 ± 0.01 Fa | 71.61 ± 0.02 Fa | 71.61 ± 0.02 Fa |
| | a* | 6.29 ± 0.01 Fc | 6.45 ± 0.03 Ec | 6.75 ± 0.02 Dc | 6.94 ± 0.02 Cc | 7.12 ± 0.01 Bc | 7.56 ± 0.02 Ac | 7.58 ± 0.02 Ac | 7.58 ± 0.01 Ac | 5.66 ± 0.01 Gc | 5.65 ± 0.03 Gc | 7.59 ± 0.01 Ac |
| | b* | 21.15 ± 0.01 Ac | 20.64 ± 0.02 Bc | 20.41 ± 0.02 Cc | 20.11 ± 0.03 Dc | 19.85 ± 0.02 Ec | 18.92 ± 0.01 Fc | 18.91 ± 0.03 Fc | 18.91 ± 0.02 Fc | 18.91 ± 0.02 Fc | 18.91 ± 0.03 Fc | 18.91 ± 0.02 Fc |
| | ΔE | 0 | 0.54 ± 0.03 Ec | 0.88 ± 0.02 Dc | 1.31 ± 0.03 Cc | 1.80 ± 0.03 Bc | 2.95 ± 0.02 Ac | 2.97 ± 0.01 Ac | 2.97 ± 0.02 Ac | 2.98 ± 0.01 Ac | 2.98 ± 0.04 Ac | 2.98 ± 0.02 Ac |
| 0.25% | |  |  |  |  |  |  |  |  |  |  |  |
| | L* | 67.56 ± 0.01 Ab | 67.46 ± 0.02 Bb | 67.39 ± 0.03 Cb | 66.23 ± 0.02 Db | 65.43 ± 0.01 Eb | 65.28 ± 0.01 Fb | 65.27 ± 0.02 Fb | 65.26 ± 0.01 Fb | 65.26 ± 0.02 Fb | 65.26 ± 0.01 Fb | 65.26 ± 0.02 Fb |
| | a* | 10.81 ± 0.01 Fb | 10.92 ± 0.01 Eb | 11.12 ± 0.03 Db | 11.37 ± 0.02 Cb | 11.58 ± 0.01 Bb | 11.65 ± 0.02 Ab | 11.67 ± 0.02 Ab | 11.69 ± 0.03 Ab | 11.69 ± 0.03 Ab | 11.69 ± 0.03 Ab | 11.69 ± 0.02 Ab |
| | b* | 34.79 ± 0.01 Ab | 34.35 ± 0.03 Bb | 33.12 ± 0.01 Cb | 31.85 ± 0.02 Db | 31.62 ± 0.01 Eb | 30.25 ± 0.02 Fb | 30.22 ± 0.01 Gb | 30.21 ± 0.02 Gb | 30.21 ± 0.01 Gb | 30.21 ± 0.02 Gb | 30.21 ± 0.01 Gb |
| | ΔE | 0 | 0.46 ± 0.03 Ec | 1.71 ± 0.01 Db | 3.28 ± 0.02 Cb | 3.90 ± 0.01 Bb | 5.15 ± 0.02 Ab | 5.18 ± 0.01 Ab | 5.20 ± 0.02 Ab | 5.20 ± 0.01 Ab | 5.20 ± 0.02 Ab | 5.20 ± 0.01 Ab |
| 0.35% | |  |  |  |  |  |  |  |  |  |  |  |
| | L* | 59.76 ± 0.01 Ac | 59.22 ± 0.04 Bc | 58.86 ± 0.03 Cc | 57.82 ± 0.01 Dc | 57.39 ± 0.02 Ec | 56.87 ± 0.02 Fc | 56.84 ± 0.02 Gc | 56.81 ± 0.01 Gc | 56.81 ± 0.03 Gc | 56.81 ± 0.02 Gc | 56.81 ± 0.01 Gc |
| | a* | 19.96 ± 0.01 Ga | 19.96 ± 0.03 Ga | 20.25 ± 0.02 Fa | 20.43 ± 0.01 Ea | 20.68 ± 0.03 Da | 20.88 ± 0.01 Ca | 20.96 ± 0.02 Ba | 21.05 ± 0.03 Aa | 21.06 ± 0.02 Aa | 21.08 ± 0.02 Aa | 21.08 ± 0.01 Aa |
| | b* | 45.07 ± 0.01 Aa | 44.12 ± 0.03 Ba | 41.86 ± 0.01 Ca | 41.35 ± 0.01 Ca | 39.54 ± 0.02 Da | 38.25 ± 0.02 Ea | 38.11 ± 0.02 Ea | 38.11 ± 0.03 Ea | 38.05 ± 0.01 Ea | 38.04 ± 0.02 Ea | 38.04 ± 0.02 Ea |
| | ΔE | 0 | 1.09 ± 0.03 Ea | 3.34 ± 0.01 Da | 4.22 ± 0.01 Ca | 6.06 ± 0.01 Ba | 7.46 ± 0.01 Aa | 7.61 ± 0.01 Aa | 7.63 ± 0.02 Aa | 7.68 ± 0.11 Aa | 7.70 ± 0.02 Aa | 7.70 ± 0.02 Aa |

Remarks: Data are given as mean ± SD. A difference in uppercase superscript letter indicates a significant difference ($p < 0.05$) in soaking time. A difference in lowercase letter indicates a significant concentration difference ($p < 0.05$).

**Table 5.** Apparent color and colorimetric parameters (L*, a*, and b*) of the SSEFs with different SWE concentrations and soaking times in alkaline conditions.

| Treatment | | Time (min) | | | | | | | | | | |
|---|---|---|---|---|---|---|---|---|---|---|---|---|
| | | 0 | 2 | 4 | 6 | 8 | 10 | 12 | 14 | 16 | 18 | 20 |
| 0.15% | |  |  |  |  |  |  |  |  |  |  |  |
| | L* | 73.09 ± 0.03 Aa | 72.45 ± 0.03 Ba | 70.26 ± 0.02 Ca | 65.36 ± 0.02 Da | 63.21 ± 0.02 Ea | 60.86 ± 0.02 Fa | 60.74 ± 0.02 Ga | 60.74 ± 0.02 Ga | 60.73 ± 0.02 Ga | 60.73 ± 0.01 Ga | 60.73 ± 0.01 Ga |
| | a* | 6.29 ± 0.02 Fc | 11.33 ± 0.01 Ec | 17.50 ± 0.02 Dc | 20.64 ± 0.01 Cc | 23.23 ± 0.03 Bc | 24.76 ± 0.02 Ac | 24.76 ± 0.01 Ac | 24.78 ± 0.02 Ac | 24.79 ± 0.01 Ac | 24.79 ± 0.03 Ac | 24.79 ± 0.01 Ac |
| | b* | 21.15 ± 0.01 Ac | 20.26 ± 0.03 Bc | 18.74 ± 0.02 Cc | 16.39 ± 0.01 Dc | 14.53 ± 0.02 Ec | 13.95 ± 0.02 Fc | 13.86 ± 0.02 Gc | 13.86 ± 0.02 Gc | 13.85 ± 0.02 Gc | 13.85 ± 0.02 Gc | 13.85 ± 0.02 Gc |
| | ΔE | 0 | 5.16 ± 0.01 Fa | 11.81 ± 0.02 Ec | 16.98 ± 0.02 Db | 20.70 ± 0.02 Cb | 23.29 ± 0.03 Bb | 23.38 ± 0.01 Bb | 23.40 ± 0.02 Aa | 23.42 ± 0.02 Aa | 23.42 ± 0.02 Aa | 23.42 ± 0.02 Aa |

**Table 5.** *Cont.*

| Treatment | | Time (min) | | | | | | | | | | |
|---|---|---|---|---|---|---|---|---|---|---|---|---|
| | | 0 | 2 | 4 | 6 | 8 | 10 | 12 | 14 | 16 | 18 | 20 |
| 0.25% | L* | 67.55 ± 0.01 Ab | 65.66 ± 0.03 Bb | 65.53 ± 0.02 Cb | 60.22 ± 0.01 Db | 58.96 ± 0.02 Eb | 55.96 ± 0.03 Fb | 54.03 ± 0.02 Gb | 54.03 ± 0.01 Gb | 54.02 ± 0.01 Gb | 54.02 ± 0.01 Gb | 54.02 ± 0.01 Gb |
| | a* | 10.81 ± 0.02 Gb | 12.44 ± 0.02 Fb | 20.02 ± 0.01 Eb | 25.64 ± 0.02 Db | 27.22 ± 0.01 Cb | 32.54 ± 0.01 Ab | 32.51 ± 0.03 Bb | 32.50 ± 0.02 Bb | 32.50 ± 0.02 Bb | 32.50 ± 0.01 Bb | 32.50 ± 0.01 Bb |
| | b* | 34.79 ± 0.02 Ab | 33.74 ± 0.02 Bb | 32.93 ± 0.02 Cb | 30.58 ± 0.01 Db | 28.34 ± 0.02 Eb | 26.79 ± 0.03 Fb | 26.81 ± 0.02 Fb | 26.81 ± 0.02 FGb | 26.82 ± 0.01 FGb | 26.83 ± 0.01 FGb | 26.83 ± 0.02 Gb |
| | ΔE | 0 | 2.71 ± 0.03 Fc | 10.22 ± 0.01 Eb | 17.07 ± 0.02 Da | 19.62 ± 0.01 Cc | 25.90 ± 0.03 Bb | 26.79 ± 0.02 Aa | 26.78 ± 0.01 Aa | 26.78 ± 0.02 Aa | 26.78 ± 0.01 Aa | 26.78 ± 0.01 Aa |
| 0.35% | L* | 59.75 ± 0.01 Ac | 57.23 ± 0.02 Bc | 53.71 ± 0.02 Cc | 52.65 ± 0.03 Dc | 51.47 ± 0.01 Ec | 51.44 ± 0.02 EFc | 51.44 ± 0.02 EFc | 51.41 ± 0.02 FGc | 51.40 ± 0.02 Gc | 51.40 ± 0.01 Gc | 51.40 ± 0.01 Gc |
| | a* | 19.96 ± 0.01 Ha | 22.56 ± 0.01 Ga | 26.11 ± 0.03 Fa | 28.96 ± 0.02 Ea | 39.32 ± 0.01 Da | 42.39 ± 0.02 Aa | 42.25 ± 0.02 Ba | 42.25 ± 0.03 Ba | 42.23 ± 0.03 BCa | 42.21 ± 0.03 Ca | 42.21 ± 0.02 Ca |
| | b* | 45.07 ± 0.01 Aa | 42.24 ± 0.01 Ba | 40.69 ± 0.02 Ca | 38.22 ± 0.01 Da | 35.34 ± 0.01 Ea | 32.75 ± 0.01 Fa | 32.72 ± 0.01 Ga | 32.71 ± 0.02 Ga | 32.71 ± 0.02 Ga | 32.71 ± 0.02 Ga | 32.71 ± 0.02 Ga |
| | ΔE | 0 | 4.60 ± 0.01 Eb | 9.67 ± 0.01 Db | 13.35 ± 0.03 Cc | 23.20 ± 0.01 Ba | 26.91 ± 0.02 Aa | 26.80 ± 0.03 Aa | 26.82 ± 0.02 Aa | 26.80 ± 0.01 Aa | 26.79 ± 0.02 Aa | 26.79 ± 0.01 Aa |

Remarks: Data are given as mean ± SD. A difference in uppercase superscript letter indicates a significant difference ($p < 0.05$) in soaking time. A difference in lowercase letter indicates a significant concentration difference ($p < 0.05$).

In neutral conditions (Table 4), the L* of the SSEF with 0.15% SWE also exhibited ($p < 0.05$) significantly higher values (71.61–73.06) than the film with 0.25% (65.26–67.56) and 0.35% (56.81–59.76) SWE. Conversely, the a* and b* values of the SSEF containing the highest concentration of SWE showed significantly ($p < 0.05$) higher values (19.96–21.08 and 38.04–45.07, respectively) than the SSEFs with 0.25% (10.81–11.69 and 30.21–34.79, respectively) and 0.15% (6.29–7.59 and 18.91–21.15, respectively). The total color difference (ΔE) of the SSEFs with the addition of 0.15%, 0.25%, and 0.35% SWE increased significantly (0.54–2.98, 0.46–5.20, and 1.09–7.70, respectively)

The L* of the SSEF soaked in alkaline conditions with 0.15% SWE also exhibited ($p < 0.05$) significantly higher values (60.73–73.09) than the films with 0.25% (54.02–67.55) and 0.35% (51.40–59.75) SWE. In contrast, the a* and b* values of the SSEF with the highest SWE concentration showed significantly ($p < 0.05$) higher values (19.96–42.21 and 32.71–45.07, respectively) than the SSEFs with 0.25% (10.81–32.50 and 26.83–34.79, respectively) and 0.15% (6.29–24.79 and 13.85–21.15, respectively). The total color difference (ΔE) of the SSEFs with the addition of 0.15%, 0.25%, and 0.35% SWE increased significantly (5.16–23.42, 2.71–26.78, and 4.60–26.79, respectively).

## 4. Discussion

### 4.1. Physical Properties

The material's strength and the edible films' ability to retain the integrity of packed foods are determined by their mechanical properties [38]. The increases in the concentration of the SWE increased the thickness of the edible film. According to reports, natural films made for food packaging range in thickness from 0.05 to 0.20 mm [39]. The results showed that the thickness of the films was within the acceptable range. Although the casting solutions had the same weight, this thickness variation can also be related to the varied film-drying kinetics, which affect the resulting thickness and structure, as previously noted in the literature [40–42].

Conversely, increasing the SWE concentration decreased the transparency of the film. The transparency of the film offers details about the size distribution of the particles in the matrix [43]. Based on the results, the lowest TS value was obtained by the SSEF with the highest concentration of SWE. The SWE may be responsible for this phenomenon by acting as a plasticizer in the matrix of the film, resulting in a decrease in the TS of the film. By reducing the intermolecular connections between nearby molecules in the polymer

network, increases in the flexibility and chain mobility of the film can be achieved, but the mechanical strength is weakened as a result [44].

In contrast to the TS, the SWE was added in the lowest amounts to provide the lowest EAB values. This process may be brought on by an increase in the transfer of starch- and flour-polymer chains, which leads extract chemicals and starch chains to form new bonds in place of the original polymer-chain linkages. When several hydrophilic components were introduced to the matrix, the polymer system was disturbed and the chain continuities decreased, which led to poor rupture resistance [44–46].

### 4.2. Color Values of SSEFs in Different Conditions

### 4.2.1. Color Values at Different pH

The increases in the concentration of the SWE under the same pH conditions decreased the L* value of the film, while the a* and b* values increased. Considering the pH value of the color of the SSEF, the increases in the SWE concentration contributed to the increasing darkness, redness, and yellowness of the SSFE. As shown by the results of the color measurement, the values of the a*, which indicates the sample's redness, were increased by the increasing pH values. Moreover, the yellowness, indicated by the b* value, decreased with the increasing pH values. The brazilein appeared yellow when the environment was acidic; the color changed to red as the pH rose to an alkaline condition. The lowering L* value also reflected the increased darkness caused by increase in the pH into the alkaline region. The protonation and deprotonation of the hydroxyl (OH) group of polyphenolic compounds (such as anthocyanins and other flavonoids), which frequently occurred upon the shifts in the pH, resulted in alterations in the molecular structure of the brazilein and caused changes in the color value [20].

### 4.2.2. Color Values at Different Soaking Times in Different Conditions

The increase in the SWE concentration increased the* value of the film, while the L* and b* values decreased. The prolonged soaking duration was also shown to affect the color values of the SSEF significantly ($p < 0.05$). The increased soaking duration in the neutral conditions increased the a* value while decreasing the L* and b* values. Unlike the neutral condition, the acidic and alkaline environments resulted in similar results. The L* value increased with the increase in the SWE concentration, while the a* and b* values decreased. Extended soaking times were also shown to affect the color values of the SSEF significantly ($p < 0.05$). The prolonged soaking durations in the acidic and alkaline conditions increased the a* value while decreasing the L* and b* values.

As can be seen from the results, the gap in the color values between the soaking periods became increasingly comprehensive with the increases in the pH of the soaking condition. The largest gap was observed with the SSEF soaked in alkaline conditions. According to [20], brazilein exhibits stronger stability characteristics at lower pH values and reduced stability at higher pH values. These color variations are caused by the protonation and deprotonation of the hydroxyl (OH) group in brazilein. Similar results were noted for other polyphenolic pigments discovered by different researchers. According to some studies [47–50], the anthocyanins' most prevalent form in acidic conditions (pH 3) is the flavylium cation, which is more stable than the forms that exist at higher pH (such as quinoidal base, carbinol pseudo base, and chalcone). Because phenolic chemicals produce a quinone intermediate that is immediately harmed by oxidation, their stability decreases at higher pH levels [20]. Anthocyanin also plays a role in the color changes in SSEF. Anthocyanins are sensitive to pH fluctuations, since they lose color above 3.0. At pH levels below 3, anthocyanins primarily reside in the form of highly stable red flavylium cations. The rapid hydration of the flavylium cation causes the colored carbinol pseudo base to be produced when the pH rises from 4 to 5. At pH 6–7, a neutral quinoidal base (purple to violet in color) results from the deprotonation of the flavylium cation; at pH 7–8, an anionic quinoidal base is produced (blue color) [51–55].

## 5. Conclusions

It can be concluded that edible films made from surimi with the addition of SWE can be used as bio-based color sensors sensitive to pH changes. According to the results, different concentrations of the SWE significantly ($p < 0.05$) affected the physical properties of the film. The pH, soaking time, and soaking conditions also significantly ($p < 0.05$) affected the color values of the film. With the increases in the pH values, the L* and b* values decreased, while the a* value increased. Based on the evaluation of the SSEF with the different soaking conditions at prolonged soaking durations, the color changes in the film in the acidic conditions were more stable than those in the neutral and alkaline conditions. In contrast, the acidic and alkaline environments resulted in similar results: increases in the L* value and decreases in the a* and b* values were correlated with the stability of the bioactive compound exhibited in the sappan-wood extract. Further works need to be performed to strengthen the color change in the indicator and the effect on the application of SSEF to food products, along with its microbial and sensory qualities.

**Author Contributions:** Conceptualization, I.R. and J.; methodology, I.R. and E.W.; software, I.R.; validation, I.R., J. and E.W.; formal analysis, I.R.; investigation, I.R.; resources, I.R., J. and E.W.; data curation, I.R.; writing—original draft preparation, I.R.; writing—review and editing, I.R.; visualization, I.R.; supervision, I.R., J. and E.W. All authors have read and agreed to the published version of the manuscript.

**Funding:** Universitas Padjadjaran through the International Open Access Program (IOAP). The authors thank the Ministry of Research, Technology, and Higher Education of the Republic of Indonesia for the Domestic Postgraduate Education Scholarships (BPPDN) 2019 (B/67/D.D3/KD.02.00/2019).

**Institutional Review Board Statement:** Not applicable.

**Informed Consent Statement:** Not applicable.

**Data Availability Statement:** Not applicable.

**Acknowledgments:** The authors thank the Faculty of Fisheries and Marine Science for supporting laboratory equipment for this research.

**Conflicts of Interest:** The authors declare no conflict of interest.

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
