# Peer review of "Designing Bio-Based Color Sensor from Myofibrillar-Protein-Based Edible Film Incorporated with Sappan Wood (Caesalpinia sappan L.) Extract for Smart Food Packaging"

_applsci, doi:10.3390/app13148205_

Round 1

Reviewer 1 Report

This work is well written. However, for intelligent indicators, subtle change in colour from one pH to another pH value is not desirable especially for food safety purposes. As colour change is only notable at extreme pH, food will be inedible and spoiled for detection/colour change to occur. This could be highlighted in the manuscript as part of improvement/future work.  The authors could also include control (without SWE) data  in table 1 and table 2. 

Fair

Author Response

Dear Reviewer,

Thank you for the opportunity to re-submit our manuscript:

Manuscript ID: applsci-2424715
Type of manuscript: Article
Title: Development of Bio-Based Color Sensor from Myofibrillar Protein–Based Edible Film Incorporated with Sappan Wood (Caesalpinia sappan L.) Extract for Smart Food Packaging
Authors: Iis Rostini *, Junianto Junianto, Endang Warsiki
Received: 15 May 2023
E-mails: iis.rostini@unpad.ac.id, junianto@unpad.ac.id,
endangwarsiki@apps.ipb.ac.id
Submitted to section: Food Science and Technology,
https://www.mdpi.com/journal/applsci/sections/food_science_and_technology
Biodegradable and Edible Films for Food Packaging Applications
https://www.mdpi.com/journal/applsci/special_issues/89344CYP4V

We thank the editor and reviewers for their thorough reading of our manuscript and their comments and suggestions that helped us to improve the manuscript. As indicated below, we have tried to do our best to respond to all the points raised. As per suggestion of one reviewer, the title was revised to "Designing of Bio-Based Color Sensor from Myofibrillar Protein–Based Edible Film Incorporated with Sappan Wood (Caesalpinia sappan L.) Extract for Smart Food Packaging".

Sincerely yours,

Iis Rostini (iis.rostini @unpad.ac.id)

Doctoral Program of Agriculture Science, Faculty of Agriculture, Universitas Padjadjaran, Sumedang, Indonesia 45363

TEL: +62-811-9625-556

Reviewer 2 Report

This study summarizes preliminary data on the Sappon wood extract doped surimi films. No thorough characterization tests were performed on the developed films. Discussion on the obtained results is not sufficient. When the color change of Sappon wood extract is examined, visual color difference is obtained above pH 10, which is very high and  no use in food spoilage detection in general.

1. The sentence starting with ‘According to the variables that can be controlled, intelligent sensor-based packaging materials can be divided into time-temperature..’ is very long and, thereby, hard to follow. Please rephrase this sentence.

2. On page 2, when a comparison is made between myofibrillar protein-based films and known protein films, it is unclear what is referred to as those known protein films. Please elaborate here.

3. Please provide some information regarding the sappan wood extract in the Material and Method Section.  

4. Please elaborate on how the tensile strength and elongation at break measurements were performed in the Material and Method Section.

5. On page 4, check the tensile data range  given in the following sentence The determination of the TS and the elongation at break of the SSEF revealed that the values of these properties were in the

range of 1.70 - 10.15 MPa and 12.68 - 15.70%, respectively.

5. Table 1 clearly shows that tensile strength is decreased with the increase in SWE concentration. Thus the following sentence is completely wrong; Based on the results, the highest TS value was obtained by the SSEF with the highest concentration of SWE. The opposite is correct according to the data given in Table 1. Besides, explain properly why tensile strength is decreased with the increase in the SWE concentration in the Discussion section. What type of intermolecular interaction is expected in the film polymeric matrix?

6. The color change of Sappan Wood Extract is not meaningful in the tested pH range. Visual color change occurs above pH 10, which is not used in food spoilage and packaging applications.

Extensive editing of English language required. Some sentences are very long and hard to follow. 

Author Response

Dear Reviewer,

Thank you for the opportunity to re-submit our manuscript:

Manuscript ID: applsci-2424715
Type of manuscript: Article
Title: Development of Bio-Based Color Sensor from Myofibrillar Protein–Based Edible Film Incorporated with Sappan Wood (Caesalpinia sappan L.) Extract for Smart Food Packaging
Authors: Iis Rostini *, Junianto Junianto, Endang Warsiki
Received: 15 May 2023
E-mails: iis.rostini@unpad.ac.id, junianto@unpad.ac.id,
endangwarsiki@apps.ipb.ac.id
Submitted to section: Food Science and Technology,
https://www.mdpi.com/journal/applsci/sections/food_science_and_technology
Biodegradable and Edible Films for Food Packaging Applications
https://www.mdpi.com/journal/applsci/special_issues/89344CYP4V

We thank the editor and reviewers for their thorough reading of our manuscript and their comments and suggestions that helped us to improve the manuscript. As indicated below, we have tried to do our best to respond to all the points raised. As per suggestion of one reviewer, the title was revised to "Designing of Bio-Based Color Sensor from Myofibrillar Protein–Based Edible Film Incorporated with Sappan Wood (Caesalpinia sappan L.) Extract for Smart Food Packaging.

Sincerely yours,

Iis Rostini (iis.rostini @unpad.ac.id)

Doctoral Program of Agriculture Science, Faculty of Agriculture, Universitas Padjadjaran, Sumedang, Indonesia 45363

TEL: +62-811-9625-556

Reviewer 3 Report

1. Please improve the abstract to increase the understanding of the reader

2. Please discuss concisely previous study regarding the biobased color sensor

3. Please standardise the use of US or UK english of manuscript

Please improve the English quality of manuscript

Author Response

(The authors gave the same response as above.)

Reviewer 4 Report

This is an interesting topic that should be considered in Applied Science. However, the writing skills of the authors are poor hence hinder the merits of the manuscript. The manuscript has a potential if the authors could improve it by letting a native speak and an experienced food scientist to edit the entire work before resubmitting for further consideration. Having said this, I recommend major revision before reconsideration. Please see below for specific suggestions and recommendations.

Title: Please replace “Development of” with “Designing Bio-Based…………..”

Abstract

“This study was done to develop bio-based color sensor from surimi-based

color sensor incorporated with sappan wood extract (SWE) for smart food packaging.” …this basically the title the authors replicated here it doesn’t speak about the aim of the study. Please revise

The SWE with different concentration (0.15%, 0.25%, and 0.35%) was incorporated to the myofibrillar proteinbased

edible film. The sappan wood-surimi edible film (SSEF) was subjected to physical properties

analysis and the color changes at different pH values and soaking time at different condition were

evaluated. Based on the results, different concentration of the SWE significantly (p<0.05) affected

the physical properties of the film. With the increasing of pH values, the darkness, redness, and

blueness of the film was increased. Based on the evaluation of the SSEF with different soaking condition,

the color changes of the film in acidic condition were more stable than in neutral and alkaline

condition. The results from this study showed that SSEF have the possibilities to be used as a smart

food packaging possessing the capabilities to act as color sensor due to its sensitivity to the changes

in pH condition of the product.”…………….please this abstract needs revision and grammar improvement.

Introduction

The development of new packaging solutions is a result of how packaging technology

develops in response to consumer expectations for food products that are fresh, safe,

and of an exceptional quality………sorry but this hard to follow. What is the authors intended message here?

The introduction needs to be looked at again and improve as it doesn’t flow or make sense in the present form.

Materials and Methods

Raw materials……..please the authors are advice to read published papers and see how to write this section.

The other sections need to be thoroughly looked at please.

…. All obtained results were analyzed and reported as mean } SD (standard deviation)…..of how many replicates? Please include this information beneath all the tables

Results

The thickness of the SSEFs was in the range of 0.17 - 0.22 mm…..should be “The

thickness of the SSEFs ranged between 0.17 - 0.22 mm………… Please the entire manuscript required and extensive English editing to make sense.

Also please include the p values in all the table 

This manuscript needs serious English editing before further consideration. 

Author Response

(The authors gave the same response as above.)

Round 2

Reviewer 3 Report

all comments are followed and corrected

Reviewer 4 Report

The authors have improved the manuscript and i recommend accepting in the present form 

the grammar is fine now but still needs some checks